# Asymptotic expansions as control variates for deep solvers to fully-coupled forward-backward stochastic differential equations

**Makoto Naito**[1☉], **Taiga Saito**[2☉], **Akihiko Takahashi**[3☉], **Kohta Takehara**[1☉*]

**1** Tokyo Metropolitan University, Graduate School of Management, Tokyo, Japan, **2** Senshu University, Graduate School of Commerce, Tokyo, Japan, **3** The University of Tokyo, Graduate School of Economics, Tokyo, Japan

☉ These authors contributed equally to this work.

\* fin.tk.house@gmail.com

**Data availability statement:** All Python codes we use for computation are available on Github: https://github.com/Makot0922/Python-Code

## Abstract

Coupled forward-backward stochastic differential equations (FBSDEs) are closely related to financially important issues such as optimal investment. However, it is well known that obtaining solutions is challenging, even when employing numerical methods. In this paper, we propose new methods that combine an algorithm recently developed for coupled FBSDEs and an asymptotic expansion approach to those FBSDEs as control variates for learning of the neural networks. The proposed method is demonstrated to perform better than the original algorithm in numerical examples, including one with a financial implication. The results show that the proposed method exhibits not only faster convergence but also greater stability in computation.

## Introduction

Over the past few decades, there has been a notable increase in interest in backward stochastic differential equations (BSDEs) among both practitioners and academic researchers. It is well known that solving BSDEs is closely related to stochastic control problem such as portfolio optimization in finance. In contrast to traditional forward stochastic differential equations (FSDEs), these are stochastic equations with boundary conditions at a future time point $T > 0$. Let $(\Omega, \mathcal{F}, \{\mathcal{F}_t\}_{t\geq 0}, \mathbb{P})$ be a filtered probability space satisfying usual conditions. BSDEs are then typically formulated as

$$dY_t = -f(t, Y_t, Z_t, \omega)dt + Z_t dw_t; \quad Y_T = V \tag{1}$$

where $V$ is a $\mathcal{F}_T$-mesurable $\mathbb{R}^m$-valued random variable, $f: [0, \infty) \times \mathbb{R}^m \times \mathbb{R}^{m \times d} \times \Omega \to \mathbb{R}^m$ and $w$ is a $d$-dimensional Wiener process, or in its integral form,

$$Y_t = V + \int_t^T f(s, Y_s, Z_s, \omega)ds - \int_t^T Z_s dw_s. \tag{2}$$

**Funding:** The author(s) received no specific funding for this work.

**Competing interests:** The authors have declared that no competing interests exist.

A pair of $(Y,Z)$, $\mathbb{R}^m$-valued and $\mathbb{R}^{m \times d}$-valued stochastic processes respectively, is called a solution to the BSDE (1) or (2).

A forward-backward stochastic differential equation (FBSDE) is an equation in which $V$ and/or $f$, which are often called the "driver" of $Y$, depend on $X$, a solution of another FSDEs, as in

$$X_t = X_0 + \int_0^t b(s, X_s, \omega)ds + \int_0^t \sigma(s, X_s, \omega)dw_s, \tag{3}$$

$$Y_t = g(X_T, \omega) + \int_t^T f(s, X_s, Y_s, Z_s, \omega)ds - \int_t^T Z_s dw_s \tag{4}$$

where $X_0 \in \mathbb{R}^n$, $b : [0, \infty) \times \mathbb{R}^n \times \Omega \to \mathbb{R}^n$, $\sigma : [0, \infty) \times \mathbb{R}^n \times \Omega \to \mathbb{R}^{n \times d}$; $g : \mathbb{R}^n \times \Omega \to \mathbb{R}^m$ satisfies $V = g(X_T, \omega)$.

Moreover, if the functions $b$ and/or $\sigma$ depend on the solution $(Y,Z)$ to the BSDE (2), as in

$$X_t = X_0 + \int_0^t b(s, X_s, Y_s, Z_s, \omega)ds + \int_0^t \sigma(s, X_s, Y_s, Z_s, \omega)dw_s, \tag{5}$$

$$Y_t = g(X_T, \omega) + \int_t^T f(s, X_s, Y_s, Z_s, \omega)ds - \int_t^T Z_s dw_s, \tag{6}$$

the system consisting of these two equations is called a coupled FBSDE. Although the functions $b, f, g$ and $\sigma$ may contain dependence on the sample path $\omega$ beyond their dependence through $X, Y$ and $Z$, we suppress $\omega$ henceforth for notational simplicity. One of sufficient conditions for the existence of the solution to this FBSDE is provided in Ji et al. [1].

The (coupled) FBSDEs often arise in financial problems such as pricing derivatives, estimating the size of credit valuation adjustments (CVAs) and funding valuation adjustments (FVAs), and deriving optimal investments. Consequently, the solution of FBSDEs is of great importance. However, with few exceptions, FBSDEs are not analytically tractable, particularly in coupled cases. Therefore, efficient numerical computation of these equations is a highly desirable objective.

In recent times, a multitude of machine learning methodologies have been employed to investigate this subject area. In particular, following the seminal works of E et al. [2] and Han et al. [3], numerous subsequent studies have used deep neural networks to construct numerical solutions with Monte Carlo simulations, which are referred to as "deep solvers" for BSDEs. Among these, [1] develops three algorithms using deep solvers to construct numerical solutions to fully-coupled FBSDEs and demonstrates the effectiveness of their techniques in several numerical experiments.

Additionally, numerous efforts have been made to enhance the efficacy of deep solvers, which includes the implementation of a methodology known as "asymptotic expansions" in FBSDEs. Asymptotic expansion approaches in finance first emerged in pricing average options (Yoshida [4], Kunitomo and Takahashi [5]) and have since been applied to a broad class of financial issues, including; derivative evaluation under stochastic interest rates (Kunitomo and Takahashi [6], Takahashi and Matsushima [7], Antonov and Misirpashaev [8], Takahashi et al. [9], Shiraya et al. [10]); pricing barrier options (Shiraya et al. [11], Shiraya et al. [12], Kato et al. [13]); optimal portfolio problems (Takahashi and Yoshida [14], Naito and Takehara [15,16]); construction of control variates for Monte Carlo simulations (Takahashi and Yoshida [17], Takahashi and Takehara [18]) and so on. For the mathematical validity of this approach, see Yoshida [4,19] and Kunitomo and Takahashi [20].

This methodology has also been applied to the field of FBSDEs: For instance, see Fujii et al. [21], Fujii and Takahashi [22–25] and Takahashi and Yamada [26,27]. For the combination of this methodology and the deep solvers applied to to uncoupled FBSDEs, Fujii et al. [28] employs the asymptotic expansions around linear drivers as control variates in conjunction with the deep solvers. Takahashi et al. [29] also employs asymptotic expansions as control variates, providing rigorous error bounds. In [15,16], the optimal investment in complete and incomplete markets is considered, respectively. Rather than deriving the expansion of the corresponding FBSDE directly, a known result of the asymptotic expansion of the optimal portfolio presented by [14] is employed as a control variate. For other works applying the asymptotic expansion methods with deep solvers to FSDEs and/or BSDEs, see Naito and Yamada [30], Iguchi et al. [31] and Takahashi and Yamada [32] and so on. Nevertheless, to the best of our knowledge, the application of the asymptotic expansion approach to deep solvers for coupled FBSDEs is none. Accordingly, this paper proposes an improvement in the efficiency of the algorithm proposed by [1] through the use of the asymptotic expansion of coupled FBS-DEs as control variates. The proposed technique is demonstrated to outperform the original algorithm in several numerical examples, including one pertaining to optimal investment strategies in incomplete markets.

The organization of this paper is as follows. The second section describes relationship between stochastic control problems with prior knowledge and FBSDEs. The third section then derives the asymptotic expansion of the target coupled FBSDE and the fourth section proposes a new algorithm which applies the expansion as control variates to the original algorithm by [1]. The subsequent section presents a series of numerical examples that illustrate the efficacy of the proposed technique. Finally, concluding remarks are stated. Some elements omitted in this paper due to space limitation are found in our full version [33].

## Stochastic control with prior knowledge and FBSDE

In this section, we introduce a stochastic control problem with prior knowledge related to solving the coupled FBSDE (5)–(6), following arguments similar to those in [1]. First, let $\mathcal{L}^2$ denote the space of all $\mathcal{F}_t$-adapted square-integrable processes, and let $\hat{u} = \{\hat{u}_t\}_{t\in[0,T]}, \hat{z} = \{\hat{z}_t\}_{t\in[0,T]}$ be elements of this space. We consider the following control problem:

$$\inf_{u,z\in\mathcal{L}^2} \mathbf{E}\left[ |Y_T^{u,\hat{u},z,\hat{z}} - g(X_T^{u,\hat{u},z,\hat{z}})|^2 + \int_0^T |Y_t^{u,\hat{u},z,\hat{z}} - (u_t + \hat{u}_t)|^2 dt \right] \tag{7}$$

where $X^{u,\hat{u},z,\hat{z}}$ and $Y^{u,\hat{u},z,\hat{z}}$ satisfy the following FSDEs

$$\begin{aligned} X_t^{u,\hat{u},z,\hat{z}} &= X_0 + \int_0^t b(s, X_s^{u,\hat{u},z,\hat{z}}, \hat{u}_s + u_s, \hat{z}_s + z_s) ds \\ &\quad + \int_0^t \sigma(s, X_s^{u,\hat{u},z,\hat{z}}, \hat{u}_s + u_s, \hat{z}_s + z_s) dw_s, \end{aligned} \tag{8}$$

$$Y_t^{u,\hat{u},z,\hat{z}} = (\hat{u}_0 + u_0) - \int_0^t f(s, X_s^{u,\hat{u},z,\hat{z}}, Y_s^{u,\hat{u},z,\hat{z}}, \hat{z}_s + z_s) ds + \int_0^t (\hat{z}_s + z_s) dw_s. \tag{9}$$

Here we know the concrete processes $\hat{u}$ and $\hat{z}$ in advance, which can be interpreted as prior knowledge for $u$ and $z$, respectively.

**Proposition 1.** *Assume that the FBSDE* (5)–(6) *has a solution* (X,Y,Z). *Then,* $u^* = \{u_t^* := Y_t - \hat{u}_t\}_{t\in[0,T]}$ *and* $z^* = \{z_t^* := Z_t - \hat{z}_t\}_{t\in[0,T]}$ *solve the control problem* (7).

*Proof*: Substituting $\hat{u} + u^* = Y$ and $\hat{z} + z^* = Z$, the processes $X^{u^*,\hat{u},z^*,\hat{z}}$ and $Y^{u^*,\hat{u},z^*,\hat{z}}$ satisfy

$$
\begin{aligned}
X_t^{u^*,\hat{u},z^*,\hat{z}} &= X_0 + \int_0^t b(s, X_s^{u^*,\hat{u},z^*,\hat{z}}, Y_s, Z_s)ds + \int_0^t \sigma(s, X_s^{u^*,\hat{u},z^*,\hat{z}}, Y_s, Z_s)dw_s, \\
Y_t^{u^*,\hat{u},z^*,\hat{z}} &= Y_0 - \int_0^t f(s, X_s^{u^*,\hat{u},z^*,\hat{z}}, Y_s^{u^*,\hat{u},z^*,\hat{z}}, Z_s)ds + \int_0^t Z_s dw_s.
\end{aligned}
$$

Clearly, $X^{u^*,\hat{u},z^*,\hat{z}} = X$ and $Y^{u^*,\hat{u},z^*,\hat{z}} = Y$ satisfy these equation with $Y_T^{u,\hat{u},z,\hat{z}} = Y_T = g(X_T) = g(X_T^{u,\hat{u},z,\hat{z}})$. This implies the optimality of $u^*$ and $z^*$ since the infimum in (7) is achieved at zero. □

Next, we define the sub-problem with neural networks as

$$
\inf_{u,z \in \mathcal{NN}} \mathbf{E}\left[ |Y_T^{u,\hat{u},z,\hat{z}} - g(X_T^{u,\hat{u},z,\hat{z}})|^2 + \int_0^T |Y_t^{u,\hat{u},z,\hat{z}} - (u_t + \hat{u}_t)|^2 dt \right] \tag{10}
$$

where $\mathcal{NN}$ is the space of all controls represented by some neural network. Then, by the Universal Approximation Theorem of deep neural networks (e.g. Calin [34, Theorem 9.5.3]), we can always find network architectures capable of approximating $(u^*, z^*)$ in Proposition 1 with $\epsilon$-precision.

We also analyze the approximation error when the neural networks are insufficiently trained. For example, assuming Lipschitz continuity of the functions $f, b$ and $\sigma$, the $L^2$-error between the optimally controlled process $Y_T = Y_T^{u^*,\hat{u},z^*,\hat{z}}$ and $Y_T^{u,\hat{u},z,\hat{z}}$ with a suboptimal control $(u,z)$ can be evaluated via Gronwall's inequality as

$$
\begin{aligned}
&\mathbf{E}\left[ |Y_T^{u^*,\hat{u},z^*,\hat{z}} - Y_T^{u,\hat{u},z,\hat{z}}|^2 \right] \\
=\ &\mathbf{E}\left[ \left| \left( (u_0^* + \hat{u}_0) - \int_0^T f(s, X_s^{u^*,\hat{u},z^*,\hat{z}}, Y_s^{u^*,\hat{u},z^*,\hat{z}}, z_s^* + \hat{z}_s)ds + \int_0^T (z_s^* + \hat{z}_s)dw_s \right) \right.\right. \\
&\left.\left. - \left( (u_0 + \hat{u}_0) - \int_0^T f(s, X_s^{u,\hat{u},z,\hat{z}}, Y_s^{u,\hat{u},z,\hat{z}}, z_s + \hat{z}_s)ds + \int_0^T (z_s + \hat{z}_s)dw_s \right) \right|^2 \right] \\
\leq\ &C\left( \mathbf{E}[|u_0^* - u_0|^2] + \int_0^T \left( \mathbf{E}[|u_s^* - u_s|^2] + \mathbf{E}[|z_s^* - z_s|^2] \right) ds \right)
\end{aligned} \tag{11}
$$

where $C > 0$ is some constant dependent on the Lipschitz coefficients and $T$.

We reformulate the notation to explicitly emphasize the dependence on prior knowledge, denoting the optimal controls as $u^*(\hat{u})$ and $z^*(\hat{z})$. Correspondingly, the suboptimal controls are expressed as $u(\hat{u}, \hat{z})$ and $z(\hat{u}, \hat{z})$, reflecting their joint dependence on both prior components through the control problem (10).

Then, from (11) we observe the followings: First, when the prior knowledge $\hat{u}$ and $\hat{z}$ closely approximate the true solution $Y$ and $Z$, the optimal control $u^*(\hat{u}) = Y - \hat{u}$ and $z^*(\hat{z}) = Z - \hat{z}$ are small. Second, consider two distinct prior pairs $(\hat{u}_1, \hat{z}_1)$ and $(\hat{u}_2, \hat{z}_2)$ where the former provides superior approximation to $(Y, Z)$ compared to the latter. To illustrate our primary motivation for using approximations based on an asymptotic expansion as prior knowledge, for instance, let $\hat{u}_1$ and $\hat{z}_1$ be the estimates obtained via the asymptotic expansion for $Y$ and $Z$. In contrast, we set $\hat{u}_2$ and $\hat{z}_2$ to zero, which corresponds to the original algorithm in [1]. In such cases, under identical training procedures the errors for suboptimal controls $|u_s^*(\hat{u}_1) - u_s(\hat{u}_1, \hat{z}_1)|$ and $|z_s^*(\hat{z}_1) - z_s(\hat{u}_1, \hat{z}_1)|$ tend to be substantially smaller than their counterparts $|u_s^*(\hat{u}_2) - u_s(\hat{u}_2, \hat{z}_2)| = |Y - u_s(0,0)|$ and $|z_s^*(\hat{z}_2) - z_s(\hat{u}_2, \hat{z}_2)| = |Z - z_s(0,0)|$, particularly during initial training phases due

to the typical random parameter initialization, which accelerates the convergence of the algorithm as observed in Numerical Example Section. Moreover, providing prior knowledge for either $Y$ or $Z$ alone may prove insufficient, as evidenced in Example 2 of that section, due to the simultaneous dependence of the controls $u$ and $z$ on $\hat{u}$ and $\hat{z}$.

## The asymptotic expansion for coupled FBSDEs

Motivated by [22], in this section we apply the asymptotic expansion approach, which is a general approximation scheme to solutions to SDEs, to the coupled FBSDE (5) and (6). First, to apply this approach we consider the following FBSDE instead of the original equations.

$$X_t^\epsilon = X_0 + \int_0^t b(s, X_s^\epsilon, \epsilon Y_s^\epsilon, \epsilon Z_s^\epsilon)ds + \epsilon \int_0^t \sigma(s, X_s^\epsilon, Y_s^\epsilon, Z_s^\epsilon)dw_s, \tag{12}$$

$$Y_t^\epsilon = g(X_T^\epsilon) + \int_t^T f(s, X_s^\epsilon, \epsilon Y_s^\epsilon, \epsilon Z_s^\epsilon)ds - \int_t^T Z_s^\epsilon dw_s \tag{13}$$

with an parameter $\epsilon \in (0, 1]$. If $\epsilon = 1$ the equations above coincide with the original ones.

Then, we approximate the solution $(X^\epsilon, Y^\epsilon, Z^\epsilon)$ to these FBSDE with their formal Taylor expansions with respect to $\epsilon$ as

$$X_t^\epsilon \sim X_t^{AE,l}, \quad Y_t^\epsilon \sim Y_t^{AE,l}, \quad Z_t^\epsilon \sim Z_t^{AE,l}, \tag{14}$$

for $l \leq 1$ where

$$X_t^{AE,l} := \sum_{n=0}^l X_t^n \frac{\epsilon^n}{n!}, \quad Y_t^{AE,l} := \sum_{n=0}^l Y_t^n \frac{\epsilon^n}{n!}, \quad Z_t^{AE,l} := \sum_{n=0}^l Z_t^n \frac{\epsilon^n}{n!},$$

and for $n \geq 1$

$$X_t^0 := \lim_{\epsilon \downarrow 0} X_t^\epsilon, \quad X_t^n := \frac{\partial^n X_t^\epsilon}{\partial \epsilon^n}\bigg|_{\epsilon=0}, \quad Y_t^0 := \lim_{\epsilon \downarrow 0} Y_t^\epsilon, \quad Y_t^n := \frac{\partial^n Y_t^\epsilon}{\partial \epsilon^n}\bigg|_{\epsilon=0},$$

$$Z_t^0 := \lim_{\epsilon \downarrow 0} Z_t^\epsilon, \quad Z_t^n := \frac{\partial^n Z_t^\epsilon}{\partial \epsilon^n}\bigg|_{\epsilon=0}.$$

In particular, we have the following concrete expression for the leading two terms, that is, the $\epsilon^0$- and $\epsilon^1$- order ones.

**Proposition 2.** *First, the $\epsilon^0$-order terms are give by*

$$X_t^0 = X_0 + \int_0^t b^0(s)ds, \quad Y_t^0 = g(X_T^0) + \int_t^T f^0(s)ds \tag{15}$$

*where $b^0(t) := b(t, X_t^0, 0, 0)$ and $f^0(t) := f(t, X_t^0, 0, 0)$, and $Z_t^0 \equiv 0$.*
*Next, the $\epsilon^1$-order terms are given by*

$$X_t^1 = \tilde{X}_t^{-1}\left(\int_0^t \tilde{X}_s \tilde{b}(s)ds + \int_0^t \tilde{X}_s \sigma^0(s)dw_s\right), \tag{16}$$

$$Y_t^1 = \Xi(t, T) + \Delta(t, T)X_t^1, \tag{17}$$

$$Z_t^1 = (Z_{ik,t}^1)_{ik}, \quad Z_{ik,t}^1 = \Delta_i(t, T)\sigma_k^0(t), \quad 1 \leq i \leq m, \quad 1 \leq k \leq d \tag{18}$$

*where $\tilde{X}_t := \exp\left(-\int_0^t \nabla_x b(t, X_t^0, 0, 0)ds\right)$, $\bar{b}(t) := \nabla_y b(t, X_t^0, 0, 0)Y_t^0$ and $\sigma^0(t) := \sigma(t, X_t^0, Y_t^0, 0)$. $\Xi(t, T), \Delta(t, T)$ are certain deterministic functions whose definition is given in Section A of [33], and $\Delta_i(t, T)$ is i-th row of $\Delta(t, T)$ and $\sigma_k^0$ is k-th column of $\sigma^0$. $\nabla_x$ and $\nabla_y$ are differential operators with respect to each element of x and y, whose concrete definitions are also given in [33].*

*Proof: See [33].* □

Obviously, the leading terms $(X_t^0, Y_t^0)$ are both deterministic processes as $b^0(t)$ and $f^0(t)$ are deterministic vector-valued functions, and the first-order terms $(X_t^1, Y_t^1)$ follow Gaussian distributions as $\tilde{X}_t, \bar{b}(t), \sigma^0(t), \Xi(t, T)$ and $\Delta(t, T)$ are all deterministic matrix-valued functions. In contrast, for the approximation of $Z_t$, $Z_t^0$ is actually zero and $Z_t^1$ is deterministic. We emphasize that, despite this simple structure, these asymptotic expansions serve as effective prior knowledge for the algorithm of [1], as confirmed in Numerical Example Section.

**Remark 1.** *In principle, the terms in higher-order expansion can be computed straightforwardly as the $\epsilon^1$-order terms. For example, $\epsilon^2$-order terms $(X_t^2, Y_t^2, Z_t^2)$ satisfy the following equations:*

$$X_t^2 = \int_0^t \left(\partial_x b^0(s)X_s^2 + \partial_x^2 b^0(s)(X_s^1)^2 + \partial_y^2 b^0(s)(Y_s^0)^2 + 2\partial_y b^0(s)Y_s^1 + 2\partial_z b^0(s)Z_s^1\right.$$
$$\left. +2\partial_x\partial_y b^0(s)X_s^1 Y_s^0\right)ds + 2\int_0^t (\partial_x \sigma^0(s)X_s^1 + \partial_y \sigma^0(s)Y_s^0)dw_s, \tag{19}$$

$$Y_t^2 = \int_t^T \left(\partial_x f^0(s)X_s^2 + \partial_x^2 f^0(s)(X_s^1)^2 + \partial_y^2 f^0(s)(Y_s^0)^2 + 2\partial_y f^0(s)Y_s^1 + 2\partial_z f^0(s)Z_s^1\right.$$
$$\left. +2\partial_x\partial_y f^0(s)X_s^1 Y_s^0\right)ds + g''(X_T^0)(X_T^1)^2 + g'(X_T^0)X_T^2 + \int_t^T Z_s^2 dw_s, \tag{20}$$

*where $\partial_\xi = \frac{\partial}{\partial\xi}$ for $\xi \in \{x, y, z\}$, $\partial_\xi^n b^0(s) := \partial_\xi^n b(s, X_s^0, 0, 0)$, $\partial_\xi^n f^0(s) := \partial_\xi^n f(s, X_s^0, 0, 0)$ and $\partial_\xi \sigma^0(s) := \partial_\xi \sigma(s, X_s^0, Y_s^0, 0)$. Here we assume that $n = m = d = 1$ to avoid complicated notation. Thanks to the decoupled structure of $(X_t^2, Y_t^2)$, namely $X^2$ depends only on $X^1, Y^1, Z^1, Z^0, Y^0$ but not on $Y^2$, this system can be solved easily. However, when the dimension of the system n,m and d increases, as in Example 1 in Numerical Examples Section, the computational burden grows substantially, making implementation challenging even at the second order. This is why we focus on the low-order expansions for control variates.*

**Remark 2.** *Instead of using the equations (12)–(13), it seems natural to start from redefining the original FBSDE (5)–(6) as*

$$X_t^\epsilon = X_0 + \int_0^t b(s, X_s^\epsilon, Y_s^\epsilon, Z_s^\epsilon)ds + \epsilon \int_0^t \sigma(s, X_s^\epsilon, Y_s^\epsilon, Z_s^\epsilon)dw_s, \tag{21}$$

$$Y_t^\epsilon = g(X_T^\epsilon) + \int_t^T f(s, X_s^\epsilon, Y_s^\epsilon, Z_s^\epsilon)ds - \int_t^T Z_s^\epsilon dw_s. \tag{22}$$

With this reformulation, the leading term $X_t^0$, $Y_t^0$ and $Z_t^0$ satisfy the following FBSDE:

$$X_t^0 = X_0 + \int_0^t b(s, X_s^0, Y_s^0, Z_s^0) ds, \qquad (23)$$

$$Y_t^0 = g(X_T^0) + \int_t^T f(s, X_s^0, Y_s^0, Z_s^0) ds - \int_t^T Z_s^0 dw_s. \qquad (24)$$

Obviously $Z_t^0 \equiv 0$ satisfies these equations, and $X_t^0$ and $Y_t^0$ are given as the solution to the coupled equations as

$$X_t^0 = X_0 + \int_0^t b(s, X_s^0, Y_s^0, 0) ds, \qquad (25)$$

$$Y_t^0 = g(X_T^0) + \int_t^T f(s, X_s^0, Y_s^0, 0) ds, \qquad (26)$$

which usually presents greater computational complexity compared to the decoupled equations (15). Furthermore, the $\epsilon^1$-order terms satisfy

$$X_t^1 = \int_0^t (\nabla_x b(s, X_s^0, Y_s^0, 0) X_s^1 + \nabla_y b(s, X_s^0, Y_s^0, 0) Y_s^1 + \nabla_z b(s, X_s^0, Y_s^0, 0) \vec{Z}_s^1) ds$$
$$+ \int_0^t \sigma^0(s) dw_s, \qquad (27)$$

$$Y_t^1 = \int_t^T (\nabla_x f(s, X_s^0, Y_s^0, 0) X_s^1 + \nabla_y f(s, X_s^0, Y_s^0, 0) Y_s^1 + \nabla_z f(s, X_s^0, Y_s^0, 0) \vec{Z}_s^1) ds$$
$$+ \nabla_x g(X_T^0) X_T^1 - \int_t^T Z_s^1 dw_s \qquad (28)$$

where $\vec{Z}_t^1 = (Z_{11,t}^1, \ldots, Z_{1d,t}^1, Z_{21,t}^1, \ldots, Z_{2d,t}^1, \ldots, Z_{d1,t}^1, \ldots, Z_{dd,t}^1)^\top$. This coupled system often requires much more computational efforts, while it admits the solution $(X^1, Y^1, Z^1)$ so that $X_t^1$ and $Y_t^1$ are jointly Gaussian distributed and $Z_t^1$ is deterministic as well as that in Proposition 2. Although there are several ways to introduce perturbations as in (21)–(22), which may provide more precise approximations in the same orders of expansions, our focus is on computational feasibility in many concrete examples as possible including high dimensional problems.

## The asymptotic expansion as control variates

In this section the asymptotic expansion derived in the previous section is combined with the algorithm of [1]. Although we develop the algorithms with combination of all the three algorithms of [1] and our asymptotic expansion, here the one for Algorithm 2 of [1] with the asymptotic expansion is displayed and the others are left in [33]. We note that the use of our asymptotic expansion as controls with Algorithm 1 and 3 in [1] is also quite effective, as shown in [33].

In particular, the asymptotic expansions $Y_t^{AE,l}$ and $Z_t^{AE,l}$ are used as control variates for the neural networks with replacement of $\phi^1$ by $\chi \times Y_t^{AE,l} + \phi^1$ and $\phi^2$ by $\chi \times Z_t^{AE,l} + \phi^2$. With setting $\chi = 1$ you obtain the proposed algorithm, while with setting $\chi = 0$ its original version is recovered, though it is also found in [1] and [33]. As discussed in the first section, this algorithm can approximate the solution with any precision when the neural networks are deep enough. Furthermore, if the approximate processes $(Y^{AE,l}, Z^{AE,l})$ are sufficiently close to the

true process $(Y,Z)$, we can expect remainders should be small and thus be easier to learn. The detailed algorithm is given as below.

**Algorithm 1** Algo. 2 in [1] (feedback control based on X) + the asymptotic expansion.

**Input:** The Wiener process $\Delta w_{t_i}$, initial parameters $\theta^0$, learning rate $\eta$, binary parameter $\chi \in \{0,1\}$; the functions $b,f,\sigma,g$ are given in (5)–(6);

**Output:** $X_T^\pi$ and the process $(Y_{t_i}^\pi)_{0 \leq i \leq N}$.

1: **for** $k = 1$ to maxstep **do**
2: **for** $m = 1$ to M **do**
3: $L^m = 0$;
4: $X_0^{k,\pi,m} = X_0$;
5: $Y_0^{k,\pi,m} = \chi \times Y_0^{AE,l} + \phi^1(X_0; \theta_0^{1,k-1})$;
6: **for** $i = 0$ to N−1 **do**
7: $u_{t_i}^{k,\pi,m} = \chi \times Y_{t_i}^{AE,l} + \phi^1(X_{t_i}^{k,\pi,m}; \theta_i^{1,k-1})$;
8: $Z_{t_i}^{k,\pi,m} = \chi \times Z_{t_i}^{AE,l} + \phi^2(X_{t_i}^{k,\pi,m}; \theta_i^{2,k-1})$;
9: $X_{t_{i+1}}^{k,\pi,m} = X_{t_i}^{k,\pi,m} + b(t_i, X_{t_i}^{k,\pi,m}, u_{t_i}^{k,\pi,m}, Z_{t_i}^{k,\pi,m})\Delta t_i + \sigma(t_i, X_{t_i}^{k,\pi,m}, u_{t_i}^{k,\pi,m}, Z_{t_i}^{k,\pi,m})\Delta w_{t_i}$;
10: $Y_{t_{i+1}}^{k,\pi,m} = Y_{t_i}^{k,\pi,m} - f(t_i, X_{t_i}^{k,\pi,m}, Y_{t_i}^{k,\pi,m}, Z_{t_i}^{k,\pi,m})\Delta t_i + Z_{t_i}^{k,\pi,m}\Delta w_{t_i}$;
11: $L^m = L^m + \frac{T}{N}|Y_{t_{i+1}}^{k,\pi,m} - u_{t_{i+1}}^{k,\pi,m}|^2$
12: **end for**
13: $L^m = L^m + |Y_T^{k,\pi,m} - g(X_T^{k,\pi,m})|^2$
14: **end for**
15: $Loss_k = \frac{1}{2M}\sum_{m=1}^M L^m$;
16: $\theta^k = \theta^{k-1} - \eta\nabla Loss_k$
17: **end for**

In the algorithm, $Y_t^{k,\pi,m}$ and $u_t^{k,\pi,m}$ are realization of $Y_t^{k,\pi}$ and $u_t^{k,\pi}$ on the $m$-th path respectively. Further, $\nabla Loss_k$ is computed by

$$\nabla Loss_k = \frac{1}{M}\sum_{m=1}^M \left[ (Y_T^{k,\pi,m} - g(X_T^{k,\pi,m}))(\nabla Y_T^{k,\pi,m} - g'(X_T^{k,\pi,m})\nabla X_T^{k,\pi,m}) \right.$$
$$\left. + \frac{T}{N}\sum_{i=0}^{N-1}(Y_{t_{i+1}}^{k,\pi,m} - u_{t_{i+1}}^{k,\pi,m})(\nabla Y_{t_{i+1}}^{k,\pi,m} - \nabla u_{t_{i+1}}^{k,\pi,m}) \right]. \tag{29}$$

This algorithm can be combined with methods such as El Mouatasim et al. [35] for even greater efficiency.

Following the recommendation by [1], we use one $n$-dim input layer, two hidden $(n + 10)$-dim layers for both networks $\phi^1$ and $\phi^2$, and a $m$-dim output layer for $\phi^1$ and a $(m \times d)$-dim output layer for $\phi^2$.

In this algorithm, the entire processes $Y^{AE,l} = \{Y_t^{AE,l}\}_{0 \leq t \leq T}$ and $Z^{AE,l} = \{Z_t^{AE,l}\}_{0 \leq t \leq T}$ are both used as the control variates for $Y$ and $Z$ respectively, where the two neural networks are applied to estimate the difference between those two approximation and corresponding true processes. Note that $Y^{AE,l}$ is stochastic even when $l = 1$, while in the other algorithms proposed in [33] employ only $Z^{AE,1}$ as deterministic controls. $Y_{t_i}^n$ can be directly obtained with an explicit expression with respect to $(X_t^n)_{n \leq l}$ and $(Y_t^n, Z_t^n)_{n \leq l-1}$ such as (17), while there is an alternative way given in [33]. The difference between the computation above and its alternative seems to be very small as shown there.

## Numerical examples

In this section, we confirm the effectiveness of the proposed method by several numerical examples for coupled FBSDEs. Due to space limitations, some of them are omitted and left in [33]. Unless otherwise stated, the parameters for the neural network are set to be as follows: A batch size is 256; a learning rate is 0.005; a number of discretization for the partition $\pi$ is 25.

Batch normalization is applied to each layer, and the Adam optimizer is employed. The networks are implemented using Python pytorch, and the code is publicly available on Github [36].

## Coupled FBSDEs which do not contain $Z$ in the forward equation for $X$

First, in this subsection we apply the proposed method to the FBSDE where the coefficients for $X$ depend only on $Y$ but not on $Z$. Concretely, the following FBSDE is considered.

**Example 1.**

$$
\begin{aligned}
X_t &= X_0 + \int_0^t b(s, X_s, Y_s)ds + \int_0^t \sigma(s, X_s, Y_s)dw_s, \\
Y_t &= g(X_T) + \int_t^T f(s, X_s, Y_s, Z_s)ds - \int_t^T Z_s^\top dw_s
\end{aligned}
$$

*where $X_0 = 0 \in \mathbb{R}^d$ and*

$$
b_i(t, x, y) = \tfrac{t}{2}\cos^2(y + x_i), \quad \sigma_{i,j}(t, x, y) = \begin{cases} \tfrac{t}{2}\sin^2(y + x_i); & i = j, \\ 0; & i \neq j, \end{cases}
$$

$$
f(t, x, y, z) = \sum_{i=1}^d z_i - \tfrac{1}{d}\left(1 + \tfrac{t}{2}\right)\sum_{i=1}^d x_i^2 - \tfrac{t}{d}\sum_{i=1}^d x_i(x_{i+1} + t)
$$

$$
- \tfrac{t^2}{d^2}\sum_{i=1}^d (x_{i+1} + t)\sin^4(y + x_i),
$$

$$
g(x) = \tfrac{1}{d}\sum_{i=1}^d x_i^2(x_{i+1} + T), \quad x_{d+1} := x_1.
$$

This can be found in [1] and the exact solution is explicitly given by $Y_t = \tfrac{1}{d}\sum_{i=1}^d X_{t,i}^2(X_{t,i+1} + t)$.

Figs 1 and 2 depict the comparison among the original method, the proposed method and the method using the asymptotic expansion alone, indicated as "original", "with AE" and "only AE", respectively, for $T = 0.1$ and $d = 100$. The results for the other algorithms are found in [33] as well as they are in the other examples below. We generate ten independent sets of paths with different random seeds, and average the results as conducted in [1]. The comparison is presented in terms of the computational time. In order to facilitate comparison, the computational time is standardized such that the time required for the original method to compute 10,000 iterations is set to one. These path generation and standardization in computational time are maintained for all subsequent figures. Note that for the method "only AE" the estimates for $Y_0$ and $Z$ are computed immediately using the explicit formulas, while the loss function is determined as the average of the values obtained over the batches employed during neural network training.

In both of the loss function and the error for $Y_0$, the proposed method with the asymptotic expansions as control variates, demonstrates a notable enhancement in performance relative to the original method even with taking computational time into consideration. Notably, the use of the asymptotic expansions not only enhances the accuracy of the estimates during the initial learning steps but also throughout the entire learning process, that is, the levels to which the loss function or the error converge. Additionally, it is observed that the proposed method exhibits slight fluctuations in its results, yet these are less significant as the order of magnitude of the error is smaller than that of the original method. In comparison to the method using the asymptotic expansion alone, the proposed method, which employs the expansion as control variates, significantly outperforms in both the size of the error for $Y_0$ and the value of the loss function.

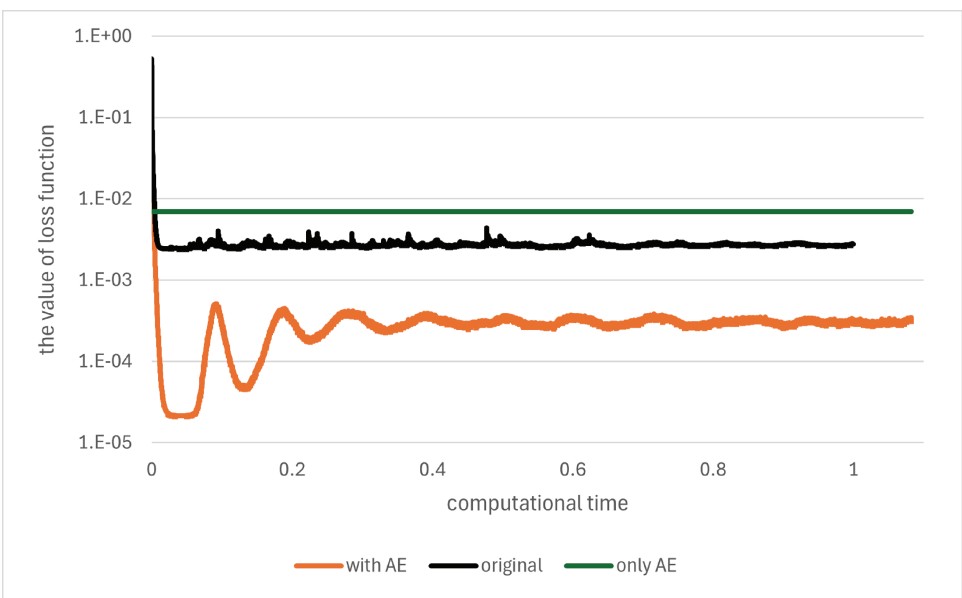

**Fig 1. Results for Example 1 by the original method (indicated as "original"), the asymptotic expansion (indicated as "only AE") and the proposed method (indicated as "with AE").** The value of the loss function is plotted against computational time on the horizontal axis: $d = 100$.

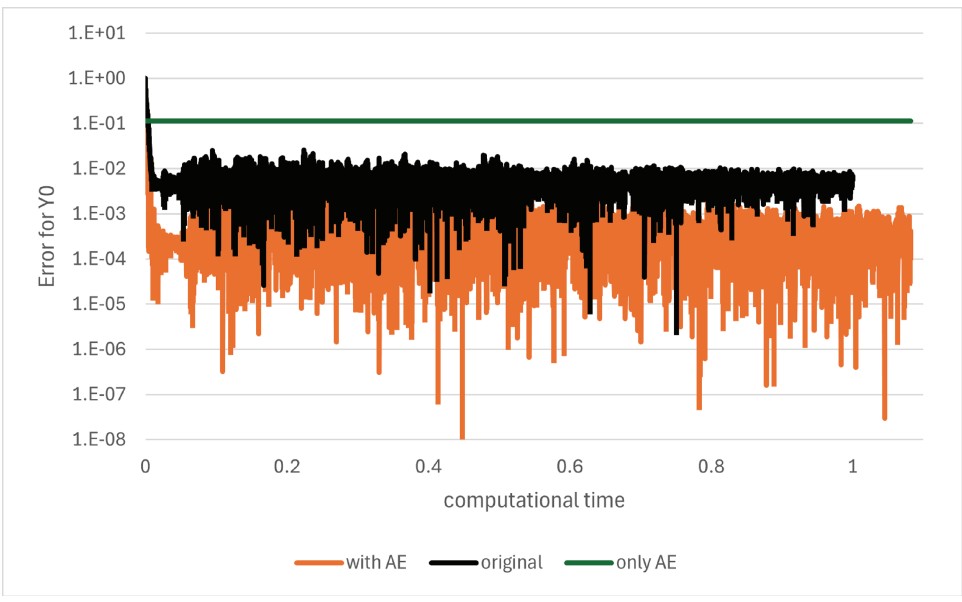

**Fig 2. Results for Example 1 by the original method (indicated as "original"), the asymptotic expansion (indicated as "only AE") and the proposed method (indicated as "with AE").** The error for the value of $Y_0$ is plotted against computational time on the horizontal axis: $d = 100$.

## Coupled FBSDEs which contains $Z$ in the equation for $X$

In this subsection, we confirm how the proposed method works in the case where the coefficients for $X$ depend both on $Y$ and $Z$. It is well known that obtaining a good numerical solution for such FBSDEs is significantly more challenging than for ones in which $X$ depends only on $Y$.

First, we see the following one-dimensional example found in [1] as with the previous one.

**Example 2.**

$$X_t = X_0 + \int_0^t b(s, X_s, Y_s, Z_s)ds + \int_0^t \sigma(s, X_s, Y_s, Z_s)dw_s,$$

$$Y_t = g(X_T) + \int_t^T f(s, X_s, Y_s, Z_s)ds - \int_t^T Z_s dw_s$$

*where $X_0 = 1$ and*

$$b(t, x, y, z) = -\tfrac{1}{2}\sin(t + x)\cos(t + x)(y^2 + z),$$
$$\sigma(t, x, y, z) = \tfrac{1}{2}\cos(t + x)(y\sin(t + x) + z + 1),$$
$$g(x) = \sin(T + x), \quad f(t, x, y, z) = yz - \cos(t + x).$$

In this case we have the exact solution $Y_t = \sin(t + X_t), Z_t = \cos^2(t + X_t)$ and $Y_0 = \sin 1 \approx 0.84147$. Following [1], we conduct comparison of the estimates obtained by applying the slightly modified asymptotic expansion to the original algorithm with those obtained by the original algorithm itself, in this example.

Figs 3 and 4 depict the comparison of the original and proposed algorithms and the method using the asymptotic expansion alone. Here, we compare the result when only the asymptotic expansions of $Y_0$ and $\{Y_t\}_{t>0}$ are employed as control variates, denoted as "with Y0&Y," with the result when the expansion of $Z$ is solely used, denoted as "with Z," and with the result when all the expansions are used in conjunction, denoted as "with AE." While incorporating the asymptotic expansions of either $Y$ or $Z$ individually as control variates yields only marginal convergence improvement, their combined implementation (denoted "with AE") achieves substantially superior performance relative to the original method. In

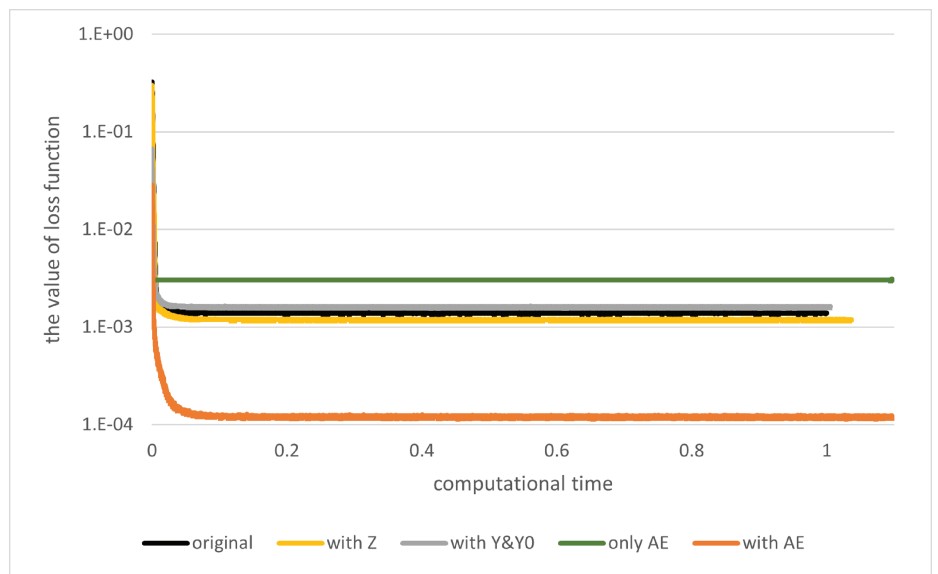

**Fig 3. Results for Example 2 by the original method (indicated as "original"), the asymptotic expansion (indicated as "only AE") and several versions of the proposed method (indicated as "Y0&Y", "with Z" and "with AE").** The value of the loss function is plotted against computational time on the horizontal axis: $d = 1$.

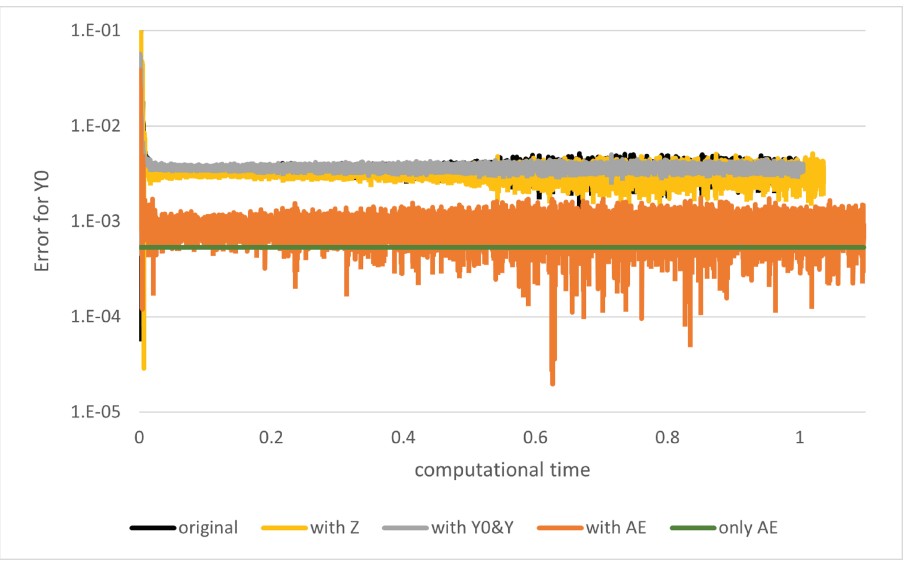

**Fig 4. Results for Example 2 by the original method (indicated as "original"), the asymptotic expansion (indicated as "only AE") and several versions of the proposed method (indicated as "Y0&Y", "with Z" and "with AE").** The error for the value of $Y_0$ is plotted against computational time on the horizontal axis: $d = 1$.

contrast, when compared to the method using the expansion alone, the error for $Y_0$ is slightly larger, which is due to a particular feature of this example. In fact, in this example the $\epsilon^0$-order terms of the expansion are given as

$$X_t^0 \;=\; X_0 + \int_0^t 0\, ds = X_0 = 1, \tag{30}$$

$$Y_t^0 \;=\; \sin(T+1) + \int_t^T (-\cos(s+1))\, ds = \sin(t+1), \tag{31}$$

and hence $Y_0^0 = \sin(1)$, which perfectly matches the true solution $Y_0 = \sin(1)$. Although including the "correction term" $Y_0^1$ introduces some error, the performance of the method using the expansion alone is still better than that of the proposed method. However, even in such cases, the value of the loss function achieved by the proposed method is much smaller than that of the method using the expansion alone.

For high-dimensional cases, [1] provides Example 4 in Section 5.4. However, we do not use this example, since it is easily shown by Ito's Lemma that any choice of $Z$ with some regularity conditions satisfies this equation, and applying our asymptotic expansion finds one of the exact solutions.

Next, we see another example found in Horst et al. [37] for coupled FBSDEs with $Z$ in the coefficient for $X$ with $d = 6$. This example is closely related to one of the most important problems in finance, namely portfolio optimization in incomplete markets.

**Example 3.** *For the forward SDEs for* $\theta_t \in \mathbb{R}^{d_1}$ *and* $H_t, X_t \in \mathbb{R}$, *we have*

$$\theta_t \;=\; \theta_0 + \int_0^t \mu_\theta(s, \theta_s)ds + \int_0^t \sigma_\theta(s, \theta_s)dw_s^{\mathcal{H}}, \tag{32}$$

$$H_t \;=\; H_0 + \int_0^t \mu_H(s, H_s)ds + \int_0^t \sigma_H(s, H_s)dw_s^{\mathcal{O}}, \tag{33}$$

$$X_t \;=\; X_0 + \frac{1}{1-\gamma}\int_0^t X_s(Z_s^{\mathcal{H}} + \theta_s)^\top \theta_s ds + \frac{1}{1-\gamma}\int_0^t X_s(Z_s^{\mathcal{H}} + \theta_s)^\top dw_s^{\mathcal{H}} \tag{34}$$

where $\mu_\theta : [0,T] \times \mathbb{R}^{d_1} \to \mathbb{R}^{d_1}, \mu_H(s,h) : [0,T] \times \mathbb{R} \to \mathbb{R}, \sigma_\theta : [0,T] \times \mathbb{R}^{d_1} \to \mathbb{R}^{d_1 \times d_1}, \sigma_H :$
$[0,T] \times \mathbb{R} \to \mathbb{R}^{d_2}$ and $w^\top = (w^{\mathcal{H},\top}, w^{\mathcal{O},\top}) = (w_1, \dots, w_{d_1}, w_{d_1+1}, \dots, w_{d_1+d_2})$ is orthogonal $(d_1 +$
$d_2)$-dimensional Wiener process.

Then, the backward SDE is given by

$$Y_t = -(1-\gamma)\ln\left(1 + \frac{\min(H_T, \bar{H})}{X_T}\right) - \frac{1}{2}\int_t^T \left(-\frac{\gamma}{1-\gamma}|Z_s^{\mathcal{H}} + \theta_s|^2 - |Z_s|^2\right)ds - \int_t^T Z_s^\top dw_s$$

where $Z_t^{\mathcal{H}} := (Z_{1,t}, \dots, Z_{d_1,t})^\top, Z_t := (Z_{1,t}, \dots, Z_{d_1,t}, Z_{d_1+1,t}, \dots, Z_{d_1+d_2,t})^\top$ and $\bar{H} \in \mathbb{R}_+$.

This fully-coupled FBSDE appears in solving the portfolio optimization problem such as

$$\sup_\pi \mathbb{E}\left[\frac{(X_T^\pi + \min(H_T, \bar{H}))^\gamma}{\gamma}\right]$$

where

$$X_t^\pi = x + \int_0^t X_s^\pi \pi_s^\top dS_s, \quad \pi_{i,t} = \frac{1}{1-\gamma}(Z_{i,t} + \theta_{i,t}) \qquad (35)$$

for $i = 1, \dots, d_1$ and the market securities $S$ are driven by $w^{\mathcal{H}}$ and the other risk from $w^{\mathcal{O}}$ is unhedgable. In contrast to the previous two examples, to our best knowledge, the exact solution to this FBSDE system is hard to obtain, while its existence is guaranteed by [37, Theorem 5.9].

Especially, we set

$$\mu_{\theta,i}(t,\theta) = \kappa(\bar{\theta}_i - \theta_i), \quad \sigma_{\theta,ij}(t,\theta) = \begin{cases} \bar{\sigma}_i, & i = j, \\ 0, & i \neq j \end{cases}$$
$$\mu_H(t,h) = \bar{\mu}_H h, \quad \sigma_H(s,h) = \bar{\sigma}_H h$$

with $d_2 = 1$ (i.e. $w^{\mathcal{O}} = w_{d_1+1}$) and $\kappa, \bar{\sigma}_i, \bar{\sigma}_H > 0, \bar{\theta}_i, \bar{\mu}_H \in \mathbb{R}$. Thus, $\theta$ is Gaussian and $H_T$ is lognormal.

Moreover, instead of expanding (33) and (34) directly, we expand their log-transformation;

$$
\begin{aligned}
h_t &= h_0 + \int_0^t (\bar{\mu}_H - \frac{\bar{\sigma}_H^2}{2})ds + \int_0^t \bar{\sigma}_H dw_s^{\mathcal{O}}, \\
x_t &= x_0 + \int_0^t \left(\frac{1}{1-\gamma}(Z_s^{\mathcal{H}} + \theta_s)^\top \theta_s - \frac{1}{2(1-\gamma)^2}|Z_s^{\mathcal{H}} + \theta_s|^2\right)ds \\
&\quad + \frac{1}{1-\gamma}\int_0^t (Z_s^{\mathcal{H}} + \theta_s)^\top dw_s^{\mathcal{H}}
\end{aligned}
$$

where $h_t := \ln H_t$ and $x_t := \ln X_t$. The idea behind this transformation is that the original processes $H_t, X_t$ seem to follow lognormal-like distributions in our setting, whereas their first-order expansions are normally distributed. It is confirmed in several numerical examples that this transformation slightly improves the performance of the proposed method, which are available upon request.

Figs 5 and 6 presents a comparison of the results for Algorithm 1 in Example 3. The parameters are set to be as follows; $\gamma = 0.5, \theta_{i,0} = \bar{\theta}_i \equiv 0.2, \bar{\sigma}_i \equiv 0.1, \kappa = 0.695; H_0 = 1, \bar{\mu}_H = 0.005, \bar{\sigma}_H = 0.1; d_1 = 5$ and $T = 1$. In this example the exact solution is not available anymore. Moreover, in practice, the value of $Y_0$ (and the maximum expected utility achieved with the

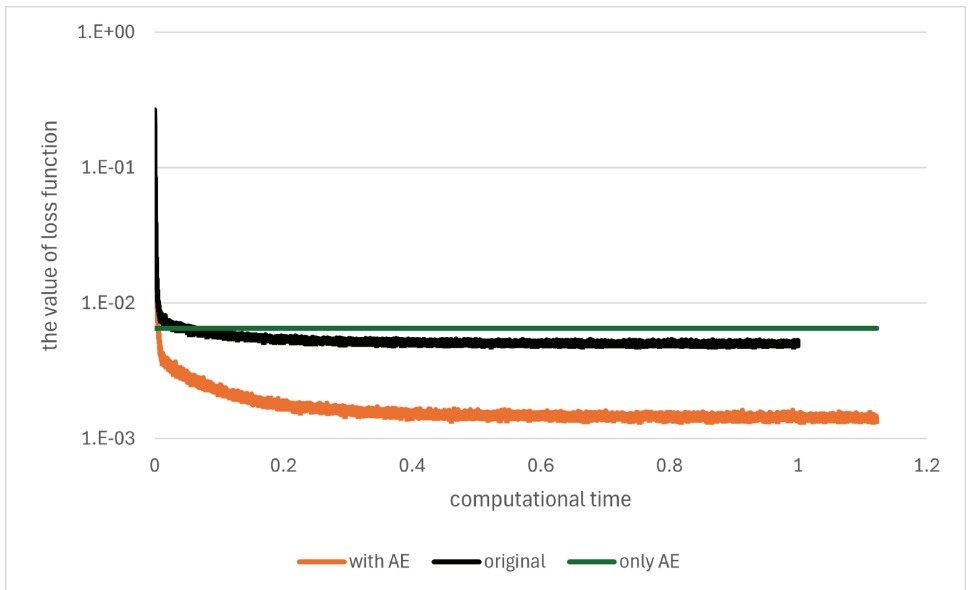

**Fig 5. Results for Example 3 by the original method (indicated as "original"), the asymptotic expansion (indicated as "only AE") and the proposed method (indicated as "with AE").** The value of the loss function is plotted against computational time on the horizontal axis: $d_1 + d_2 = 5 + 1$.

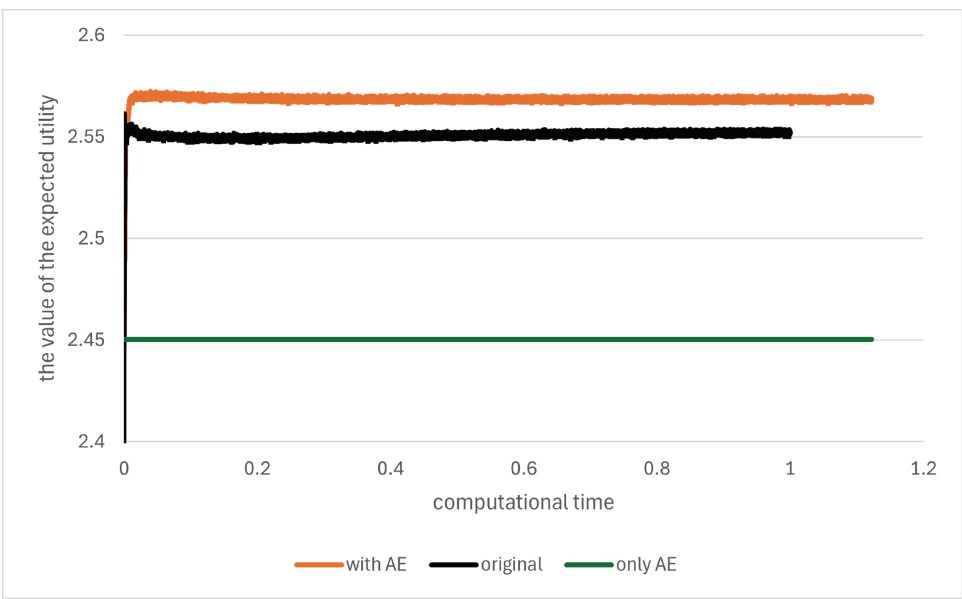

**Fig 6. Results for Example 3 by the original method (indicated as "original"), the asymptotic expansion (indicated as "only AE") and the proposed method (indicated as "with AE").** The value of the expected utility achieved with the estimated portfolio is plotted against computational time on the horizontal axis: $d_1 + d_2 = 5 + 1$.

*true* optimal portfolio) is often of secondary interest. Instead, the focus is on which portfolio offers the greatest expected utility compared to other portfolios. In this sense, the expected utility achieved with the portfolio obtained from the numerical solution of $Z$ via (35) is

estimated by the out-of-sample-path average (using $10^5$ paths for this computation) and we replace the result for $Y_0$ by that for this criterion.

As seen in the figures, the proposed method not only resulted in a notable improvement in the value of the loss function, but also in the expected utility compeared to the original method and the method using the expansion alone. Furthermore, the use of the asymptotic expansions as the prior knowledge significantly improves the stability of the computation in the following sense. In the original version of Algorithm 1, [1] proposes the random selection of the initial value for the learning process for $Y_0$. Nevertheless, if the aforementioned approach is employed, whereby the initial value is generated from a range of [–2,2] as proposed in [1], it is observed that 90 out of 100 trials fails to update the neural network within the first 100 learning steps. This phenomenon is observed consistently across a range of parameter settings, which are not reported here for brevity. In contrast, when $Y_0^{AE,1}$ is used as the input, the computation is successful in updating the network without exceptions. This stability in computation is noteworthy.

In summary, in all the examples presented in this section, the proposed method improves the efficiency of the algorithm by [1] and the method using the asymptotic expansion alone.

## Concluding remarks

In this paper, we proposed the new method which combines the algorithm proposed by [1] for coupled FBSDEs and the asymptotic expansions of those FBSDEs as the control variates for learning of the neural networks. In the examples including ones with high dimensionality or the financially important implication, it is numerically confirmed that our proposed method performed better than the original algorithm. This improvement was not only for the values of the loss functions or the errors, but for the stability of the algorithm.

For future research, we refer to the followings: First, we try to give a rigorous error bound which was not done in this paper. Second, we can incorporate higher-order terms than one in the asymptotic expansions. From the fact that the level to which the values of the loss function and the error for $Y_0$ converge was improved in Algorithm 1 with the stochastic control variate $Y^{AE,1}$, it is expected that the use of these higher-order random variables as additional control variates will further improve the efficiency of the proposed method. Finally, we are interested in other examples with financial implication such as general equilibrium in incorporate markets.

## Author contributions

**Investigation:** Makoto Naito, Taiga Saito, Akihiko Takahashi, Kohta Takehara.

**Methodology:** Taiga Saito, Akihiko Takahashi, Kohta Takehara.

**Visualization:** Kohta Takehara.

**Writing – original draft:** Akihiko Takahashi, Kohta Takehara.

**Writing – review & editing:** Makoto Naito, Taiga Saito, Akihiko Takahashi, Kohta Takehara.

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
