## [Decision Letter · Decision Letter 0]

23 Dec 2024

PONE-D-24-54964Asymptotic expansions as control variates for deep solvers to fully-coupled forward-backward stochastic differential equationsPLOS ONE

Dear Dr. Takehara,

Thank you for submitting your manuscript to PLOS ONE. After careful consideration, we feel that it has merit but does not fully meet PLOS ONE’s publication criteria as it currently stands. Therefore, we invite you to submit a revised version of the manuscript that addresses the points raised during the review process.

We look forward to receiving your revised manuscript.

Kind regards,

Viswanathan Arunachalam, Ph.D.

Academic Editor

PLOS ONE

Journal Requirements: When submitting your revision, we need you to address these additional requirements. 1. Please ensure that your manuscript meets PLOS ONE's style requirements, including those for file naming. The PLOS ONE style templates can be found at https://journals.plos.org/plosone/s/file?id=wjVg/PLOSOne_formatting_sample_main_body.pdf and https://journals.plos.org/plosone/s/file?id=ba62/PLOSOne_formatting_sample_title_authors_affiliations.pdf 2. We note that your Data Availability Statement is currently as follows: All relevant data are within the manuscript and its Supporting Information files. Please confirm at this time whether or not your submission contains all raw data required to replicate the results of your study. Authors must share the “minimal data set” for their submission. PLOS defines the minimal data set to consist of the data required to replicate all study findings reported in the article, as well as related metadata and methods (https://journals.plos.org/plosone/s/data-availability#loc-minimal-data-set-definition). For example, authors should submit the following data: - The values behind the means, standard deviations and other measures reported;- The values used to build graphs;- The points extracted from images for analysis. Authors do not need to submit their entire data set if only a portion of the data was used in the reported study. If your submission does not contain these data, please either upload them as Supporting Information files or deposit them to a stable, public repository and provide us with the relevant URLs, DOIs, or accession numbers. For a list of recommended repositories, please see https://journals.plos.org/plosone/s/recommended-repositories. If there are ethical or legal restrictions on sharing a de-identified data set, please explain them in detail (e.g., data contain potentially sensitive information, data are owned by a third-party organization, etc.) and who has imposed them (e.g., an ethics committee). Please also provide contact information for a data access committee, ethics committee, or other institutional body to which data requests may be sent. If data are owned by a third party, please indicate how others may request data access. 3. Please upload a copy of Supporting Information Figure/Table/etc. "S1 Fig. Bold the title sentence." which you refer to in your text on page 9/11.

**Additional Editor Comments:**

We have received the reports from our advisors on your manuscript “ Asymptotic expansions as control variates for deep solvers to fully-coupled forward-backward stochastic differential equations”

We invite you to revise your paper. When your revision is ready, please submit the updated manuscript and a point-by-point response.

Reviewers' comments:

Reviewer's Responses to Questions

**Comments to the Author**

1. Is the manuscript technically sound, and do the data support the conclusions?

Reviewer #1: Partly

Reviewer #2: Yes

2. Has the statistical analysis been performed appropriately and rigorously? 

Reviewer #1: Yes

Reviewer #2: Yes

3. Have the authors made all data underlying the findings in their manuscript fully available?

Reviewer #1: No

Reviewer #2: Yes

4. Is the manuscript presented in an intelligible fashion and written in standard English?

Reviewer #1: Yes

Reviewer #2: Yes

5. Review Comments to the Author

Reviewer #1: The paper titled “Asymptotic expansions as control variates for deep solvers to fully-coupled forward-backward stochastic differential equations”, submitted to PLOS ONE. The authors proposed an algorithm for learning of the neural networks (NN).

Minor Comments :

- Algorithm 1 missing !

- The notation of functions in Algo 2 such as $g$ should be cleared

- Add the computation of gradient loss in Algo 2

- Which programming language used for implementation? If Python give the information’s such as Pytorch, cuda, ..

- The architect of NN model used should be given.

- Please see this reference

El Mouatasim, A., de Cursi, J.E.S. & Ellaia, R. Stochastic perturbation of subgradient algorithm for nonconvex deep neural networks. Comp. Appl. Math. 42, 167 (2023).

Major Comments :

- Convergence analysis of the proposed algorithm should be given

- The code of algorithm should be given in open source such as repository references.

Reviewer #2: Overall, I think this is an interesting study and it is well-organized. The proposed approximation formula provides an accurate and efficient simulation method for some FBSDEs and has the potential to be applied in various situations. However, a few areas can be improved. I offer some suggestions and questions for improvement. If these suggestions are addressed and the questions are resolved, I am confident that this study will be worthy of publication in the Plos One.

Please refer to the attached file for details.

6. PLOS authors have the option to publish the peer review history of their article (what does this mean?). If published, this will include your full peer review and any attached files.

Reviewer #1: No

Reviewer #2: No

---

## [Author Response · Author response to Decision Letter 1]

7 Feb 2025

We greatly appreciate the reviewers' valuable comments, which have significantly improved the quality of our paper.

We have revised the manuscript and describe how we respond to the comments by each reviewer in separate files named "Response to reviewer 1.pdf" and "Response to reviewer 2.pdf", respectively.

Please see these files.

---

## [Decision Letter · Decision Letter 1]

12 Mar 2025

Asymptotic expansions as control variates for deep solvers to fully-coupled forward-backward stochastic differential equations

PONE-D-24-54964R1

Dear Dr. Takehara,

We’re pleased to inform you that your manuscript has been judged scientifically suitable for publication and will be formally accepted for publication once it meets all outstanding technical requirements.

Kind regards,

Viswanathan Arunachalam, Ph.D.

Academic Editor

PLOS ONE

Additional Editor Comments (optional):

Reviewers' comments:

Reviewer's Responses to Questions

**Comments to the Author**

1. If the authors have adequately addressed your comments raised in a previous round of review and you feel that this manuscript is now acceptable for publication, you may indicate that here to bypass the “Comments to the Author” section, enter your conflict of interest statement in the “Confidential to Editor” section, and submit your "Accept" recommendation.

Reviewer #1: (No Response)

Reviewer #2: All comments have been addressed

2. Is the manuscript technically sound, and do the data support the conclusions?

Reviewer #1: Yes

Reviewer #2: Yes

3. Has the statistical analysis been performed appropriately and rigorously? 

Reviewer #1: N/A

Reviewer #2: Yes

4. Have the authors made all data underlying the findings in their manuscript fully available?

Reviewer #1: Yes

Reviewer #2: Yes

5. Is the manuscript presented in an intelligible fashion and written in standard English?

Reviewer #1: Yes

Reviewer #2: Yes

6. Review Comments to the Author

Reviewer #1: (No Response)

Reviewer #2: I have carefully reviewed the revised manuscript and am pleased to inform the authors that the changes requested have been addressed appropriately. Therefore, I recommend the paper for publication in PLOS ONE.

7. PLOS authors have the option to publish the peer review history of their article (what does this mean?). If published, this will include your full peer review and any attached files.

Reviewer #1: **Yes: **Abdelkrim El Mouatasim

Reviewer #2: No

---

## [Editor Report · Acceptance letter]

PONE-D-24-54964R1

PLOS ONE

Dear Dr. Takehara,

I'm pleased to inform you that your manuscript has been deemed suitable for publication in PLOS ONE. Congratulations! Your manuscript is now being handed over to our production team.

Kind regards,

on behalf of

Dr. Viswanathan Arunachalam

Academic Editor

PLOS ONE